# All-gas-phase synthesis of UiO-66 through modulated atomic layer deposition

Kristian Blindheim Lausund[1] & Ola Nilsen[1]

Thin films of stable metal-organic frameworks (MOFs) such as UiO-66 have enormous application potential, for instance in microelectronics. However, all-gas-phase deposition techniques are currently not available for such MOFs. We here report on thin-film deposition of the thermally and chemically stable UiO-66 in an all-gas-phase process by the aid of atomic layer deposition (ALD). Sequential reactions of $ZrCl_4$ and 1,4-benzenedicarboxylic acid produce amorphous organic–inorganic hybrid films that are subsequently crystallized to the UiO-66 structure by treatment in acetic acid vapour. We also introduce a new approach to control the stoichiometry between metal clusters and organic linkers by modulation of the ALD growth with additional acetic acid pulses. An all-gas-phase synthesis technique for UiO-66 could enable implementations in microelectronics that are not compatible with solvothermal synthesis. Since this technique is ALD-based, it could also give enhanced thickness control and the possibility to coat irregular substrates with high aspect ratios.

[1] Centre for Materials Science and Nanotechnology (SMN), Department of Chemistry, University of Oslo, P.O. Box 1033 Blindern, Oslo N-0315, Norway. Correspondence and requests for materials should be addressed to O.N. (email: ola.nilsen@kjemi.uio.no).

Metal-organic frameworks (MOFs) are a class of compounds combining both inorganic and organic functionalities. These crystalline materials typically have a porous framework with porosity exceeding that of the well-known zeolites[1]. The pores in crystalline MOFs are a part of their crystal structure, which results in an accurate pore size control. Thanks to the wide choice of inorganic and organic building units, the range of possible compounds is huge and only a fraction of these have been explored, especially with respect to characterization of their physical properties.

Due to their very high porosity, MOFs are promising for a number of applications such as gas storage[2], catalysis[3], drug delivery[4], handling and destruction of toxins[5–7], and as membranes for desalination[8]. Practical routes for deposition of thin and conformal films of porous materials will lead to many new applications, particularly within membrane development or within microelectronics for use as active material in highly sensitive gas sensors based on cantilevers or as low-$\kappa$ dielectrics, as envisioned by Allendorf et al.[9].

Organic–inorganic hybrid materials have been deposited as thin films by atomic layer deposition (ALD) in a mode also known as molecular layer deposition (MLD)[10–12]. These films are typically amorphous and do not show the same properties as crystalline MOFs. ALD and MLD are techniques where two or more precursors are individually pulsed into a reaction chamber through the gas phase and allowed to react with, and saturate, the surface of a substrate. When the surface is saturated by the first precursor, excess precursor is removed by purging with an inert gas, and the process is repeated for the second precursor. A thin-film is constructed one atomic layer or one molecular layer at a time by reiterating these steps in a cyclic process[13].

To realize MOF applications within electronics and sensors, it is vital to develop synthesis routes that do not involve solvents, since these typically cause chemical contamination in the circuitry or stiction of small features. An all-gas-phase synthesis utilizing ALD/MLD of MOFs is therefore a highly suitable approach with very precise control of the amount of deposited material.

Possibilities for deposition of crystalline MOF thin films by vapour-phase techniques have recently emerged[14,15], opening the possibility of disruptive technologies due to the unique properties of these microporous crystalline materials. Suitable all-gas-phase processes are essential for enabling nanostructures of these compounds, as the stiction associated with wet-based techniques is avoided. The established MLD technique should be well suited for deposition of such microporous materials. However, sequential deposition makes it difficult to form the complex metal clusters found in MOF structures; instead giving amorphous structures without the required clusters. Even approaches using metal precursors providing suitable clusters in the gas phase[16] have required sequential wet-based hydrothermal treatment to crystallize in the desired structure.

One of the most stable MOFs is UiO-66. This zirconium-based MOF was discovered by Cavka et al.[17] in 2008. Thin films of this MOF would be particularly useful due to its stability in many chemical environments, which is why it was chosen for this work. There are very few other examples of synthesis of thin films of UiO-66 or solvent-free synthesis of bulk UiO-66 in literature, and no examples of thin-film deposition by all vapour-phase approaches. The prior examples of deposition of UiO-66 as thin films have been by electrophoretic deposition of pre-synthesized UiO-66 particles from a toluene suspension[18], electrochemical deposition[19] or by solvothermal growth[20]; and, to our knowledge, the only example of solvent-free synthesis of bulk UiO-66 was through mechanochemical synthesis[21].

We demonstrate a new approach utilizing a modulated MLD process where the metal-to-linker stoichiometry is controlled by modulating the process with an additional non-linker compound to construct the archetypical MOF material UiO-66 with high crystallinity. This approach opens up new possibilities in deposition of crystalline microporous materials with complex metal clusters. UiO-66 is deposited through application of $ZrCl_4$, and 1,4-benzenedicarboxylic acid (1,4-BDC, also known as terephthalic acid) as the organic linker.

## Results

**Deposition of hybrid films without modulation.** The $ZrCl_4 + 1,4$-BDC system was initially investigated using the *in situ* quartz-crystal microbalance (QCM) technique for a deposition temperature of 265 °C. The typical sensor response for growth using the sequence of a 4 s $ZrCl_4$ pulse, 6 s purge, 5 s 1,4-BDC pulse and 6 s purge is shown in Fig. 1. This pulsing sequence was used as a standard sequence throughout the QCM experiments, if not stated otherwise. The pulsing sequence showed self-limiting growth for both types of precursors, as can be seen in Fig. 2. The mass increase per precursor sums to 42.3% for the $ZrCl_4$ pulse and 57.7% for the 1,4-BDC precursor. This corresponds well with a relative mass increase of 178.35 g mol$^{-1}$ during the $ZrCl_4$ pulse and 241.11 g mol$^{-1}$ during the 1,4-BDC pulse, obtained from an average reaction scheme of:

$$ZrCl_4 \text{ pulse:} \tag{1}$$
$$|-(OH)_{1.5} + ZrCl_4(g) = |-O_{1.5}ZrCl_{2.5} + 1.5 \text{ HCl (g)}$$

$$1,4\text{-BDC pulse:}$$
$$|-O_{1.5}ZrCl_{2.5} + 2C_6H_4(COOH)_2(g) = |-O_{1.5}Zr(C_6H_4O_{3.25})_2(OH)_{1.5} + 2.5 \text{ HCl (g)} \tag{2}$$

The pulse/purge system was also investigated by QCM based on the growth as averaged over 16 cycles when changing the individual pulse or purge parameters of the 4-6-5-6 standard sequence for two sensors situated 5 cm apart along the direction of the gas stream (Fig. 2). This experiment was repeated twice. The reactions were self-saturating and delayed saturation of $ZrCl_4$ along the flow stream was found based on our dual-QCM sensor approach, as can be seen from the lower growth rate in the back of the chamber with pulse lengths of 0.75, 1 and 2 s; this is to a certain degree also seen for 1,4-BDC since the growth rate is lower in the back of the chamber with 0.25 and 0.5 s pulses (Fig. 2a,c). The standard pulsing scheme of 4-6-5-6 is well within ALD-type growth conditions.

The growth of the $ZrCl_4 + 1,4$-BDC system was investigated as a function of deposition temperature (Fig. 3a). A number of

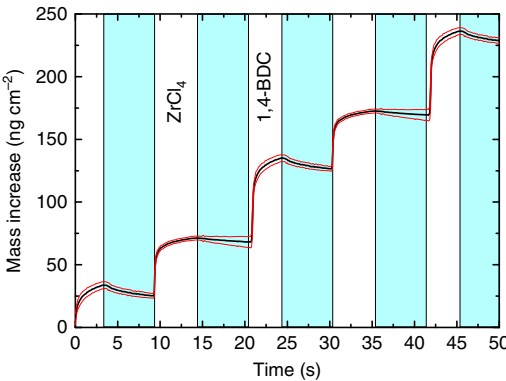

**Figure 1 | QCM characterization.** Mass gain as a function of time measured with QCM for the ALD/MLD system with alternating $ZrCl_4$ and terephthalic acid (1,4-BDC) pulses separated by inert gas ($N_2$) purges (blue). The red lines show the s.d. for the QCM data ($n = 16$).

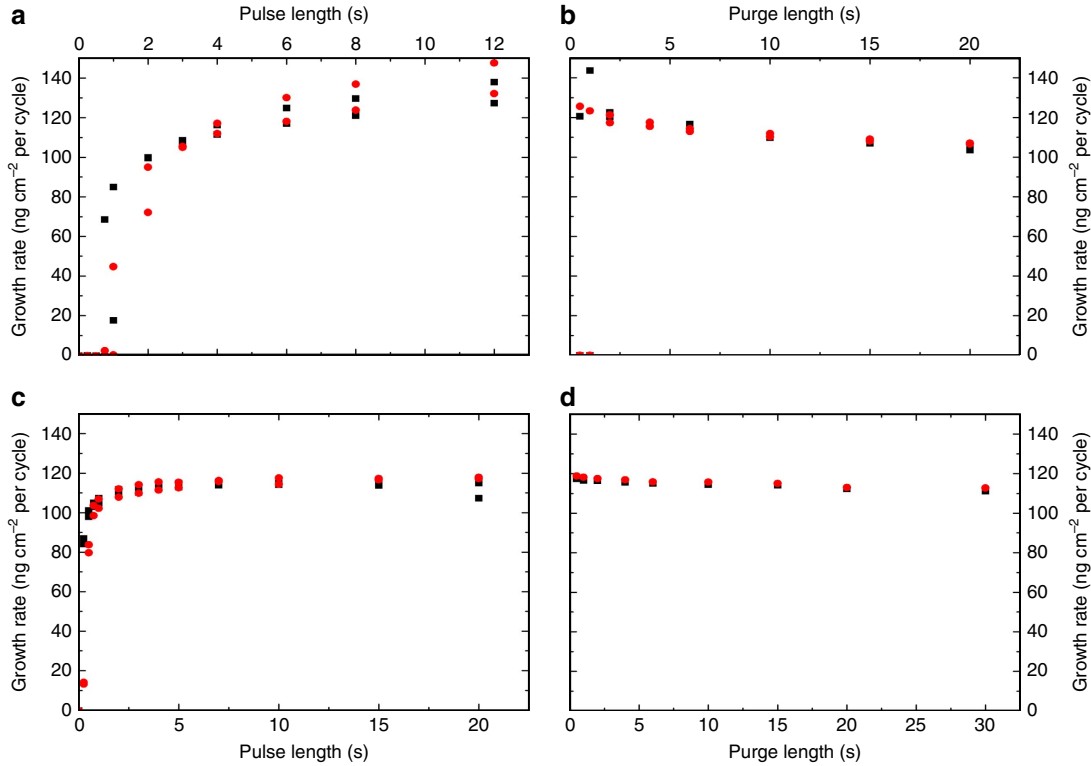

**Figure 2 | Test of self-saturated growth.** The growth rate for the system in Fig. 1 as a function of lengths of the $ZrCl_4$ pulse (**a**), the $ZrCl_4$ purge (**b**), the 1,4-BDC pulse (**c**) and the 1,4-BDC purge (**d**). Two sensors were used, one in the front of the reaction chamber (black square), and one in the back (red dot), situated ca. 5 cm apart. The experiment was repeated twice with the same sensors.

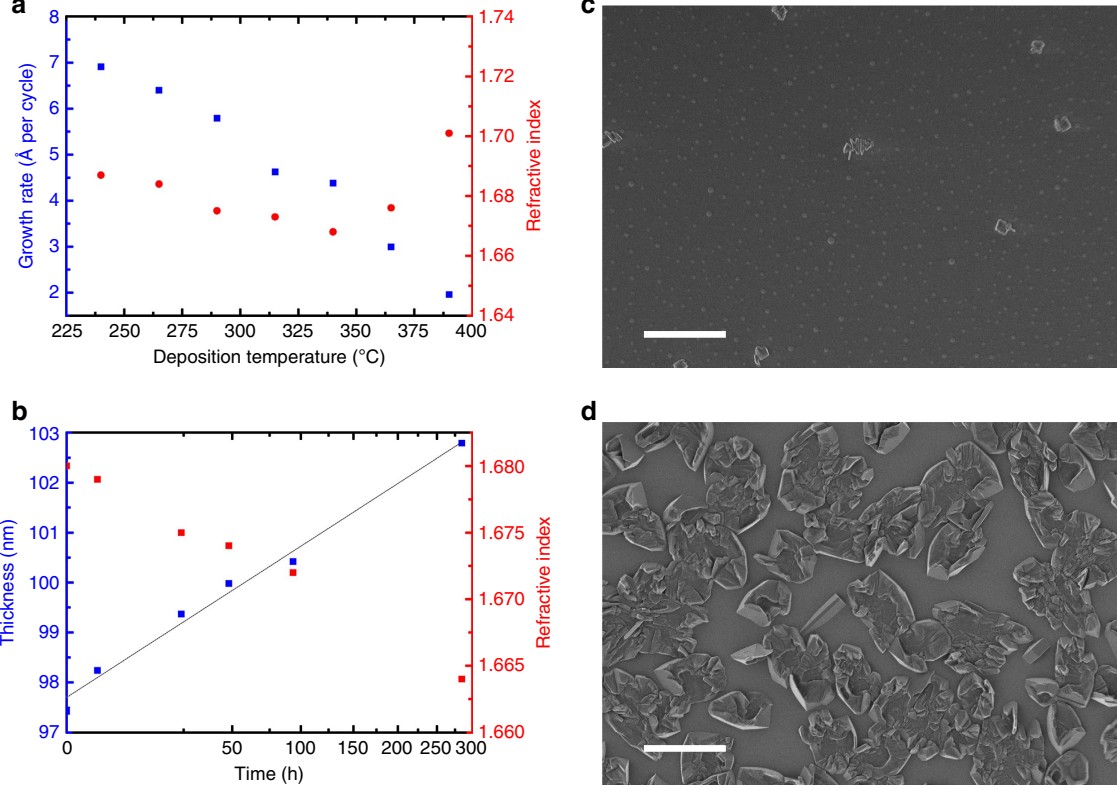

**Figure 3 | Effects of deposition temperature and storage conditions.** (**a**) Growth rate and refractive index of the $ZrCl_4 + 1,4$-BDC system as function of the deposition temperature. (**b**) Thickness and refractive index as a function of storage time with a square root dependency on the time scale. (**c,d**) show SEM images of a 130 nm film deposited at 265 °C before and after exposure to moisture, respectively. Scale bar, 2 μm.

depositions were made with 143 cycles of 4 s $ZrCl_4$ pulse, 2 s purge, 3 s 1,4-BDC pulse and 1 s purge, and increasing reactor temperature from 235 to 390 °C. The growth rate decreases with increasing temperature from 7 to 2 Å cycle$^{-1}$, and the refractive index shows a slight reduction between 235 and 335 °C, and then undergoes a significant increase.

A sample stored in air was characterized by spectroscopic ellipsometry (SE) over time to identify its environmental stability (Fig. 3b). The film thickness increased as a function of the square root of the time. Coherently, the refractive index was reduced from 1.680 to 1.664.

The increase in thickness over time is most likely due to a reaction with moisture from the air. A selection of the samples was exposed to a moist environment with a relative humidity of 70–75% at room temperature for 24 h. This resulted in an increase in thickness of ca. 20% and a drastic change in topography, as can be seen from the scanning electron microscope (SEM) images in Fig. 3d. Judged by the SEM image in Fig. 3c the as-deposited films are rather smooth. This is also confirmed by X-ray reflectometry (XRR) analysis, where we see a roughness of 0.3 nm on a 30.9-nm thick film.

Grazing incidence X-ray diffraction analyses of the films before and after exposure to a moist environment, which are presented later in the results section, indicate that some of the 1,4-BDC precursor crystallized on the surface of the films. This is most likely due to subsequent reactions between parts of the 1,4-BDC that are not optimally bonded to the Zr clusters.

**Deposition of hybrid films with water pulsing.** Within the same QCM experiment as mentioned above, four different pulsing schemes in which $H_2O$ was pulsed in addition to the two precursors were investigated to see what effect this would have on the films. One of these pulsing schemes is shown in Fig. 4. For all systems, it is evident that pulsing water causes a large mass increase that is mostly lost in the subsequent purge. This is most likely due to adsorption (followed by desorption) of water and may indicate that the film is porous to small molecules like water during deposition. This corresponds well with the porosity test performed on amorphous films, which is presented towards the end of the results section. In addition to the loss of water during the purge, there is probably also a slight reduction of the amount of 1,4-BDC in the film during the water pulse, which is concealed by the mass increase from the water uptake. This is shown by Fourier transform infrared (FTIR) spectroscopy, where we see that the peak corresponding to 1,4-BDC in monodentate

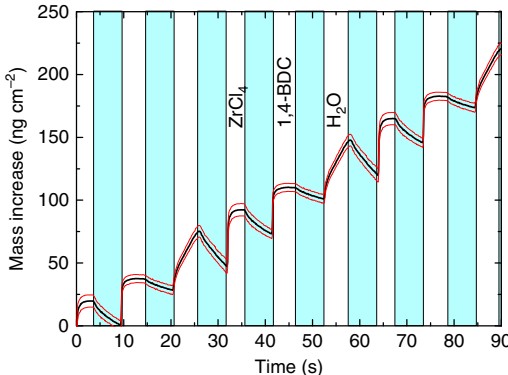

**Figure 4 | Deposition of hybrid film with $H_2O$.** Mass gain as a function of time measured with QCM for the ALD/MLD system with $ZrCl_4$, 1,4-BDC and $H_2O$. The blue panels indicate inert gas ($N_2$) purging. The red lines show the standard deviations for the QCM data ($n = 16$).

coordination is lost in a manner similar to the effect of modulation with acetic acid, which will be discussed in the next section (Supplementary Fig. 1a). Our theory that this poorly bonded 1,4-BDC is removed from the film when water is pulsed is further supported by the fact that 1,4-BDC no longer crystallizes on the surface when the films deposited with $H_2O$ pulsing are exposed to a 70–75% relative humidity over 24 h as seen by GI X-ray diffraction (Supplementary Fig. 1b). GI X-ray diffraction of a film deposited with additional water pulses also shows that the crystallinity of the films is not affected by the water pulse in that they remain amorphous as-deposited. However, the ratio of Cl to Zr is reduced from 3.2 to 0.1 wt% compared with an as-deposited film with a pulse sequence of 4 s $ZrCl_4$ pulse, 2 s purge, 3 s 1,4-BDC pulse and 1 s purge. This loss in chlorine may also contribute to a mass loss during the water pulse.

Due to the porous nature of these films and the fact that the desorption of water from the films seems to be a relatively slow process, it is possible that the film acts as a reservoir for water allowing some water to come in contact with $ZrCl_4$ during the next pulse causing unwanted reactions. This can be seen from the QCM graph in Fig. 4. The mass loss during the purge following the water pulse seems to carry over to the purges following the $ZrCl_4$ and 1,4-BDC pulses indicating that water is still leaving the film while $ZrCl_4$ is pulsed. Because of this we have chosen not to go further with this approach.

**Deposition of hybrid films with acetic acid modulation.** As mentioned above, GI X-ray diffraction analyses indicate that some of the 1,4-BDC precursor crystallized on the surface of samples deposited without modulation when exposed to a relative humidity of 70–75% over 24 h (Fig. 5). An attempt was made to reduce this excess of 1,4-BDC in the film by introducing additional pulses of acetic acid after 1,4-BDC in the deposition process. Acetic acid has previously been used as a modulator in MOF synthesis, and the hypothesis is that it may replace some of the 1,4-BDC that is not optimally bonded to the surface during its pulse[20,22]. The growth dynamics were investigated by QCM, which shows a notable reduction in mass increase when acetic acid is pulsed (Fig. 5a). Adding the acetic acid pulse does not affect the overall growth rate in terms of mass increase per cycle, as compared with the growth in Fig. 1. Both systems show an overall growth rate of ca. 100.8 ng cm$^{-2}$ cycle$^{-1}$. The mass increase for the $ZrCl_4$ is increased by 55% as compared with the process without acetic acid. By assuming that no acetic acid is included in the film during growth, as supported by the FTIR characterization presented in the end of the 'Results' section, the overall growth scheme assumes a 66% mass increase for the $ZrCl_4$ pulse and the remaining 34% during the 1,4-BDC + acetic acid pulse. This mass variation cannot be described by the reaction scheme in equation (1) and (2), which assumes an overall stoichiometry of $Zr(1,4\text{-BDC})_2$, while UiO-66 has a 1:1 stoichiometry between Zr and 1,4-BDC using $Zr_6O_4(OH)_4^{12+}$ clusters bonded to 12 –COO$^-$ functional groups. On this basis it is possible to sketch a reaction scheme according to:

$$ZrCl_4\ \text{pulse:} \tag{3}$$
$$| - (OH)_{1.75} + ZrCl_4(g) = | - O_{1.75}ZrCl_{2.25} + 1.75\ HCl\ (g)$$

$$1,4\text{-BDC pulse:}$$
$$| - O_{1.75}ZrCl_{2.25} + C_6H_4(COOH)_2(g) = | - O_{1.75}Zr(C_6H_4O_{2.25})(OH)_{1.75} + 2.25 HCl\ (g) \tag{4}$$

With a relative mass increase of 169.23 g mol$^{-1}$ and 84.10 g mol$^{-1}$ for the $ZrCl_4$ and 1,4-BDC pulses, respectively. This reaction scheme does not explain or take into account formation of the Zr clusters or the occurrence of the additional

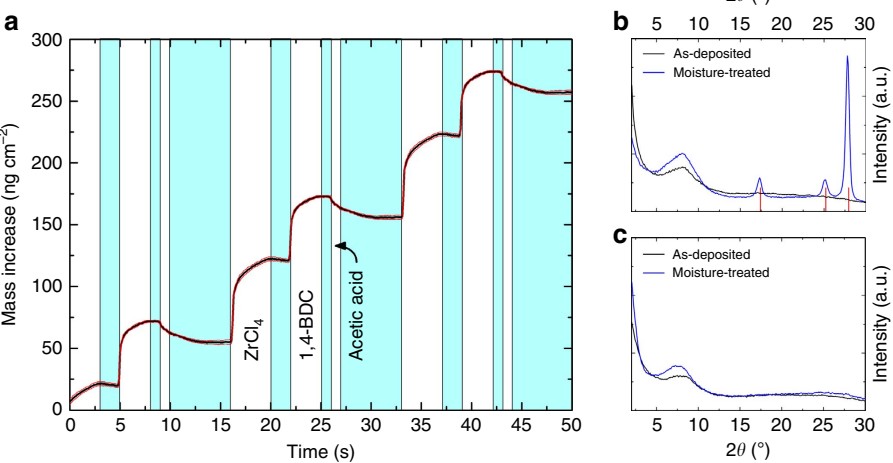

**Figure 5 | Effects of acetic acid modulation on the ALD process. (a)** QCM measurements showing mass increase (and loss) as the precursors and the modulator are pulsed. The blue panels indicate inert gas ($N_2$) purging. The red lines show the standard deviations for the QCM data ($n = 16$). (**b,c**) show GI X-ray diffractograms for the films that were deposited without and with acetic acid modulation, respectively. The black lines show the as-deposited samples, and the blue lines are the moisture-treated samples. Positions for reflections of the crystalline 1,4-BDC phase of more than 10% intensity are given by the red vertical lines in **b** (ref. 23).

$O^{2-}$ or $OH^-$. We can at this stage only assume that they are added through the subsequent reactions with air and acetic acid in the sealed autoclave or during handling before this treatment. This remains to be investigated with more dedicated techniques.

The films that were deposited with additional pulses of acetic acid were also exposed to moisture after deposition. The GI X-ray diffractogram of the moisture-treated sample no longer shows reflections corresponding to crystals of 1,4-BDC (Fig. 5c). This indicates that there is probably no longer an excess of 1,4-BDC in the films and that a more optimal growth is obtained when acetic acid is used.

**Crystallization and characterization of the films.** Several different attempts were made to induce crystallization of the deposited films. One approach was treatment under solvothermal conditions after deposition. Water and dimethylformamide were used as solvents, and different configurations were tested where the film was either submerged in the solvent or supported above it to avoid direct contact. These solvothermal treatments showed a varying degree of crystallinity but they all lacked strong XRD reflections at 2θ below 10 degrees, which would indicate crystalline UiO-66.

In addition to the solvothermal treatments, a film made with acetic acid modulation was placed in an autoclave with $\sim 0.1$ ml of acetic acid, sealed and heated to 160 °C for 24 h, essentially heating the film in acetic acid vapour (Fig. 6a).

This treatment resulted in a film with a very similar GI X-ray diffractogram to the ref. 17 for UiO-66 (Fig. 7). The thickness of the film increased from 229 nm as-deposited to $\sim 500$ nm as can be seen from the cross-section SEM image in the inset in Fig. 6b. The treatment also roughened the surface giving a root mean squared roughness of $\sim 45$ nm, as measured by atomic force microscopy (AFM). This roughness prevented XRR measurements on the treated samples (Fig. 6b,c).

A similar autoclave treatment without acetic acid in the autoclave was also performed on a sample deposited without acetic acid modulation. The resulting film was promising, as the GI X-ray diffractogram showed reflections corresponding to crystalline UiO-66, albeit with lower intensity than when acetic acid was added in the autoclave. This sample was deposited

without acetic modulation, so 1,4-BDC also crystallized on the surface.

Since XRR measurements could not be used to determine the density of the post-deposition-treated samples, a porosity test was done using QCM measurements. Two QCM-crystals were coated with 500 cycles of ZrCl4, 1,4-BDC and acetic acid for modulation, which should give a film thickness of $\sim 230$ nm. One of these crystals was then treated in the autoclave as described above. The QCM-crystals with as-deposited and post-deposition-treated films were exposed to a 5-s long water vapour pulse in the ALD reactor to determine the water uptake. The test was done at room temperature with a base pressure of $\sim 5$ mbar $N_2$. The same procedure was also done to two uncoated crystals as a control. The results show a response over 200 times higher from the post-deposition-treated sample compared with the uncoated crystals. The as-deposited sample also shows a high response, which is delayed as compared with the post-deposition-treated sample (Fig. 8). Given a sufficiently long water pulse, both samples show similar saturation values (Supplementary Fig. 2a); and with a sufficiently long purge, the water uptake is completely lost (Supplementary Fig. 2b). This indicates that both the amorphous (as-deposited sample) and the crystalline (post deposition treated sample) are porous, but that the pores in the treated sample are much more accessible. By comparing the mass increase during the water pulse (found from Supplementary Fig. 2a) with the total mass gain during the deposition (49,594 ng cm$^{-2}$) of the post-deposition-treated film we see that the water uptake in the film is $\sim 1.9$ wt%. This is at a relative humidity in the ALD chamber of $\sim 7\%$ (found by measuring a 2.2 mbar pressure increase in the chamber during the water pulse, which corresponds to the partial pressure of water, and converting this to relative humidity). For comparison, Schoenecker et al.[24] present a water uptake in UiO-66 of $\sim 2$ wt% with a relative humidity in this range. FTIR measurements were done of a modulated, as-deposited sample before and after exposure to water, to ensure that the water pulse during the porosity test does not affect the composition of the film, but no change was seen (Supplementary Fig. 3).

FTIR spectroscopy was performed on samples deposited with and without acetic acid modulation, and a modulated sample that in addition was heated in an autoclave in acetic acid vapour

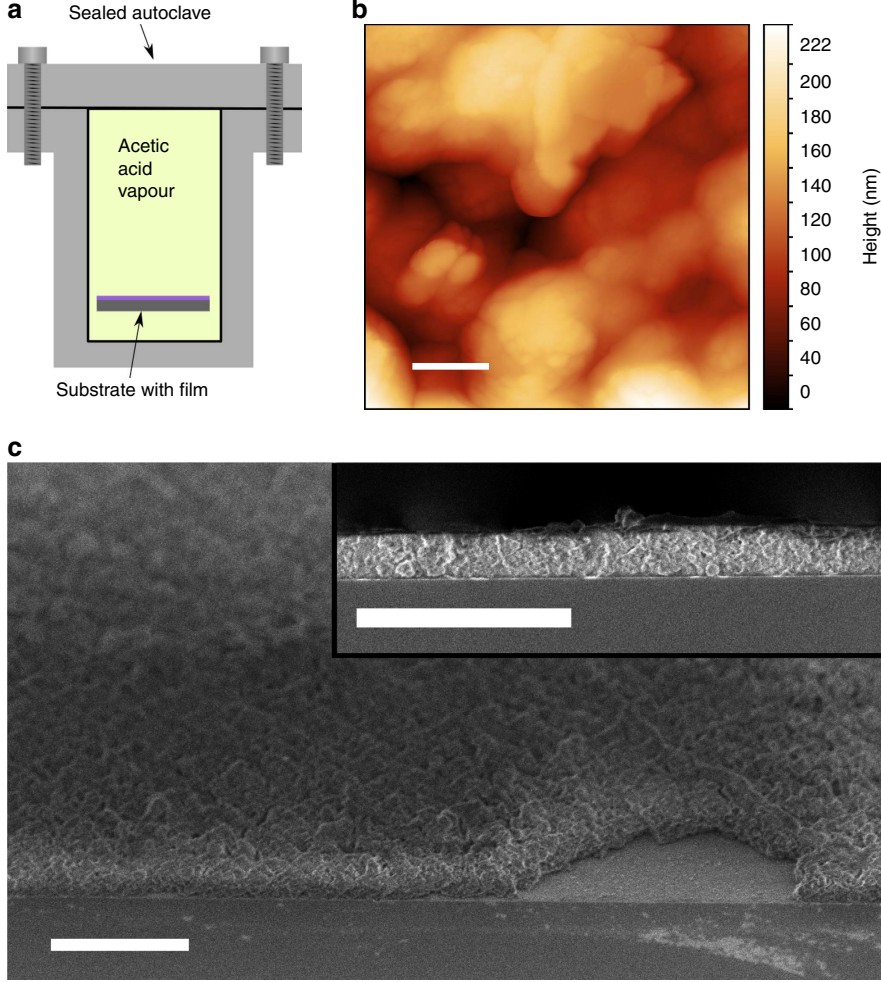

**Figure 6 | Post-deposition crystallization.** (**a**) Experimental setup for heat treatment of the films in acetic acid vapour. (**b**) AFM image of the Zr-1,4-BDC film after treatment in acetic acid vapour. (**c**) Cross-section SEM images of the same surface viewed at 45° and 90° angles. Scale bar, 0.2 μm (**b**), 2 μm (**c**).

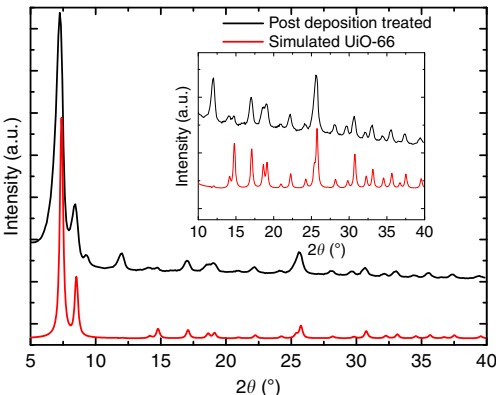

**Figure 7 | GI X-ray diffraction on crystallized film.** GI X-ray diffractogram for a film that was heated to 160 °C in acetic acid vapour in a sealed autoclave for 24 h (black) and a simulated powder diffractogram for desolvated UiO-66 (red); the inset shows the same diffractograms zoomed in on peaks with 2θ between 10 and 40 degrees.

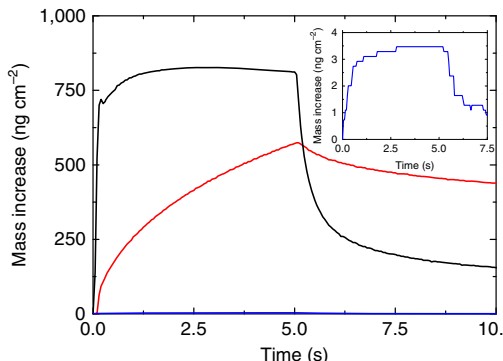

**Figure 8 | Porosity test.** Porosity test performed by monitoring the water uptake by QCM-crystals during a 5 s water pulse. The test was performed in the ALD reactor at room temperature with a base pressure of 5 mbar $N_2$. Red is the as-deposited sample (ca. 230 nm), black is the post-deposition-treated sample (ca. 500 nm), and blue is the average of the two uncoated QCM-crystals as a control. The inset shows the control enlarged.

(Fig. 9d). According to Verpoort et al.[25], the wave number separations between the most prominent peaks around $1,400 \, cm^{-1}$ and $1,500–1,700 \, cm^{-1}$ can reveal information on the coordination modes of 1,4-BDC to zirconium (Fig. 9a–c). These two peaks correspond to the symmetric and asymmetric stretches of the carboxylate group, respectively. The splitting is larger than $200 \, cm^{-1}$ when the 1,4-BDC molecule acts as a monodentate ligand, and in the range of $50–150 \, cm^{-1}$ if it acts as a bidentate ligand. A splitting of $130–200 \, cm^{-1}$ is expected for bridging ligands (Fig. 9). The amount of splitting observed by

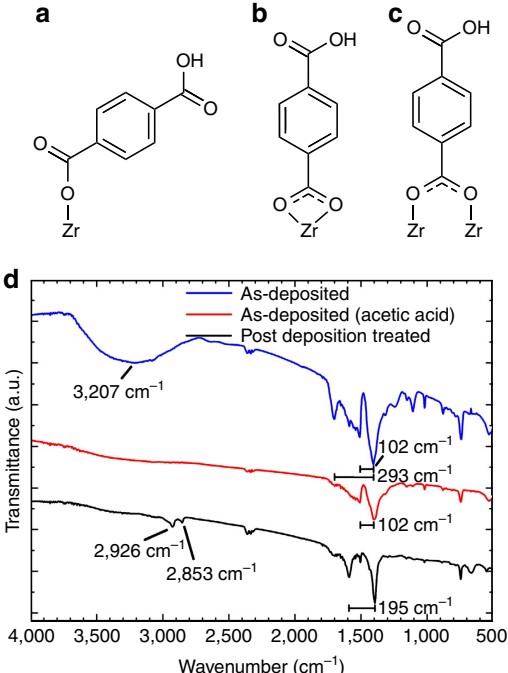

**Figure 9 | Infrared spectroscopy.** (**a**–**c**) show the possible coordination modes of 1,4-BDC to zirconium. **a** is monodentate, **b** is bidentate and **c** is bridging. (**d**) The FTIR spectra for as-deposited samples with and without acetic acid modulation and a post-deposition-treated modulated sample. The splitting between the major peaks and values of selected peaks are noted.

FTIR is shown in Fig. 9d. This shows that acetic acid modulation removes monodentate 1,4-BDC molecules, leaving only the ones with a bidentate coordination. It can also be seen that the post-deposition treatment changes the coordination from bidentate to bridging, which agrees with the UiO-66 structure.

## Discussion

The in situ QCM investigations of the ALD/MLD sequence with $ZrCl_4$ and 1,4-BDC as precursors show a self-terminating growth manner for the pure hybrid film (Fig. 2). The growth rate is $\sim$6–7 Å cycle$^{-1}$ with a reaction temperature of 265 °C, which is notably higher than normal ALD-processes, but within what is expected for an ALD/MLD process. The hybrid films show very little surface roughness as-deposited, which is as expected for growth of amorphous films (Fig. 3c). When exposed to moisture, the thickness of the films increases along with a considerable change in topography (Fig. 3b–d). During this process, pure 1,4-BDC crystallizes on the surface as shown by the GI X-ray diffractograms in Fig. 5b. Such crystallization of 1,4-BDC was prevented when additional pulses of acetic acid were introduced after the 1,4-BDC pulses. The overall growth rates for these two pulsing schemes resulted in identical values in terms of mass per area, although the relative mass change between $ZrCl_4$ and 1,4-BDC indicates a clear change in stoichiometry towards more equiatomic composition. The physical growth rates for the pulsing schemes without and with acetic acid were 8.3 Å cycle$^{-1}$ and 4.6 Å cycle$^{-1}$, respectively, indicating an increase in overall density when modulation is introduced. This has also been confirmed by XRR and can be understood by the FTIR analysis of the films proving a transition of the bonding scheme from both monodentate and bidentate ligands for films deposited without acetic acid, to only bidentate coordination when acetic acid is

used. It is possible that purely bidentate coordination yields straight columns of zirconium and 1,4-BDC linkers that orientate in a more dense manner than for a combination of bidentate and monodentate coordination, where an angled coordination between 1,4-BDC and zirconium is expected. From both the GI X-ray diffractograms and the FTIR results we can see that UiO-66 is not nucleated until it is treated with acetic acid vapour in an autoclave.

The films were crystallized by heating to 160 °C for 24 h in a sealed autoclave with $\sim$0.1 ml of acetic acid. The resulting GI X-ray diffractograms show an almost identical pattern to the X-ray diffractograms for UiO-66. The coordination also changed from bidentate to bridging, which corresponds to the coordination environment in the UiO-66 structure. This may indicate that the addition of acetic acid during deposition aids the overall composition, while proper crystallization of the UiO-66 phase does not occur before higher concentrations of acetic acid are used in the autoclave treatment. The FTIR spectrum for the as-deposited film also shows a broad peak at 3,207 cm$^{-1}$, which is possibly due to OH groups on monodentate 1,4-BDC molecules that are poorly bonded and have only coordinated to Zr on one side leaving the other carboxylic group protonated. We also see two peaks at 2,926 and 2,853 cm$^{-1}$ for the post-deposition-treated sample. This may indicate that there is some acetic acid left in the structure after the autoclave treatment. These peaks are not observed in the sample deposited with acetic acid, which indicates that no acetic acid is left in the structure after the ALD/MLD deposition.

In summary, an all-gas-phase deposition technique for porous, crystalline UiO-66 has been developed through a combination of modulated ALD and additional exposure to acetic acid vapour. The films deposited without acetic acid modulation show an excess of 1,4-BDC, but when acetic acid pulsing is used during deposition this no longer is the case. The films are amorphous as-deposited but crystallize to the UiO-66 structure on treatment in acetic acid vapour. This synthesis method has substantial advantages compared with normal MOF synthesis since thin films of UiO-66 now can be implemented in microelectronics otherwise not compatible with wet-based processing. The importance of this was also exemplified by Stassen et al.[14], who developed a synthesis route for the MOF ZIF-8 by allowing dense zinc oxide to react with vapours of the organic linker molecule. UiO-66 has superb thermal and chemical stabilities compared with other MOF materials, and is thus ideal both as a model material and for implementation in real applications. To the best of our knowledge, no all-gas-phase synthesis technique for UiO-66 thin films has been reported before this work. This method benefits from all the advantages of ALD, for example, great thickness control even on irregular substrates with high aspect ratios. This is also a good example of a modulator being used directly in the ALD sequence for MOF synthesis. Previously, inhibitors have been used in the ALD sequence to reduce and distribute the amount of precursor embedded as dopant material[26,27]. Our approach leads to a technique for controlling the composition of MLD materials to form crystalline MOF structures, which has, until now, been difficult. This could enable a wider selection of MOF materials to be deposited by all-gas-phase approaches in the future.

## Methods

**Atomic/molecular layer deposition.** The ALD/MLD depositions were performed in an F-120 Sat-type ALD reactor (ASM Microchemistry Ltd) using $ZrCl_4$ (MERCK Schuchardt OHG >98%) and 1,4-BDC(MERCK Schuchardt OHG ≥98%) as precursors. Type 2 water and acetic acid (MERCK KGaA 100%) were also used as co-reactants in selected experiments. The carrier and purging gas was $N_2$ separated from air in a nitrogen generator (Schmidlin UHPN3001 $N_2$ purifier, >99.999% $N_2$ + Ar purity). A total flow of ca. 250 sccm (standard cubic

cm min$^{-1}$) of N$_2$ was used throughout the experiments, leading to a background pressure of ca. 5 mBar. The vapourization temperatures for ZrCl$_4$ and 1,4-BDC were set to 165 and 220 °C, respectively, based on previous investigations[11,28]. Water and acetic acid were kept at room temperature in external containers. The films were deposited on as-received, pre-cleaned Si(001) substrates of 2 × 2 cm$^2$. Their native oxide thickness was measured by SE before deposition and taken into account when the film thickness was determined.

**Spectroscopic ellipsometry.** SE data were collected using a J.A. Woollam alpha-SE spectroscopic ellipsometer using a wavelength range of 390–900 nm, and modelled to a Cauchy-function using the CompleteEASE software package to determine the thicknesses and refractive indices (at $\lambda = 632.8$ nm) of the films.

**In situ QCM.** In situ QCM analyses were performed using two 6 MHz AT cut quartz crystals mounted ca. 5 cm apart on a home-made holder to monitor the mass increase during the deposition and uncover possible delayed saturations through the reaction chamber. The signals were recorded using a Maxtek TM-400 and processed by averaging over 16 consecutive cycles. The signal from the QCM analysis was converted from $-\Delta Hz$ to ng cm$^{-2}$ by calibrating the sensitivity of each sensor with the thickness and density data measured by XRR of films deposited separately from the QCM measurements. Variations in sensitivity of the sensors throughout the QCM experiment were also corrected for by using a standard sequence repeatedly throughout the experiment and adjusting all results based on variations in this standard. To ensure a reliable response from the QCM-crystals, the temperature was stabilised for 3 h before any experiments were performed.

**Post-deposition treatment.** Some of the films were exposed to moisture for 24 h in a chamber which was held at a relative humidity of 70–75% by storing a saturated NaCl solution in the chamber. Crystallization of the films was performed by heating the samples to 160 °C for 24 h in a sealed autoclave with an internal volume of 35 ml with $\sim$0.1 ml of acetic acid added.

**X-ray characterizations.** GI X-ray diffraction and XRR analyses were performed using a PANalytical Empyrean diffractometer, equipped with a Cu Kα source powered at 45 kV/40 mA ($\lambda = 1.5406$ Å), a parallel beam X-ray mirror and a proportional point detector (PW 3011/20). For GI X-ray diffraction the incident angle was $\omega = 0.30°$, while for XRR analysis $2\theta$ was scanned from 0.08° to 6°. The XRR results were analysed using the X'Pert Reflectivity software provided by PANalytical. X-ray fluorescence was performed on a PANalytical Axios mAX minerals instrument, and analysed using the Stratos software.

**Infrared spectroscopy.** FTIR spectroscopy was performed using a Bruker VERTEX 80 FTIR spectrometer in transmission mode. A spectrum obtained from an uncoated silicon substrate was used as background.

**Electron microscopy.** SEM images were obtained using a HITACHI SU8230 SEM with a cold cathode field emission type electron gun. The working distance was $\sim$2 mm, and the acceleration voltage was typically 1 kV with a beam current of 10 μA.

**Atomic force microscopy.** AFM and roughness measurements were performed using a Park Systems XE-70 AFM equipped with a standard NCHR tip. Data were analysed using the Gwyddion 2.43 SPM visualization tool[29].

**Simulation.** The simulation of the XRD diffractogram for UiO-66 was done using Mercury[30] based on the data from the paper by Valenzano et al.[31].

**Data availability.** The data that support the findings of this study are available from the corresponding author on request.

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

## Acknowledgements

We thank the strategic research environment Diatech@UiO at the University of Oslo for funding. K.B.L. is also thankful to the co-supervisors of this project Elsa Lundanes, Carl Henrik Gørbitz and Steven Wilson for their ideas and comments, Leva Momtazi and Greig Shearer for help with FTIR measurements, David Wragg for proof reading and to other

colleagues at the University of Oslo for their aid. We also like to thank the Norwegian national infrastructure for X-ray diffraction and scattering (RECX), and the Department of Geosciences at the University of Oslo for use of the X-ray fluorescence equipment.

## Author contributions

The experiments were planned by both authors. All depositions, post deposition treatments and characterisations were performed by K.B.L. The manuscript was written by both authors.

## Additional information

**Competing financial interests:** The authors declare no competing financial interests.

