## [Peer Review File · Nature Communications]

Reviewers' comments:

Reviewer #1 (Remarks to the Author):

The authors report a new all-vapor method to deposit the MOF material UiO-66. While I do think the work has great potential, I cannot recommend the work for publication in its current form. I would like to invite the authors to address the following issues:

- As a general comment, the manuscript is not well-structured and contains too many spelling and grammar errors to be acceptable at this level.
- The authors repeatedly claim the context of microelectronics. It seems a bit 'too easy' to make such a general claim. What are the promising applications of the material UiO-66 in this context? This point should be made very clear. I urge the authors to describe their vision of what 'microelectronics' would be enabled by the described process.
- The authors state that solvents 'typically cause corrosion or chemical contamination in the circuitry'. I do not agree at all. Solvents do not cause corrosion. On the contrary, the reactants used ($ZrCl_4$) and products formed (HCl) would likely destroy any conductive contacts that would be present in a microelectronic system. This is a critical point that should be addressed.
- The authors state: "Possibilities for deposition of dense MOF thin films by vapour phase techniques have recently emerged" and insert ref 12 and 13. I find the term 'dense' extremely confusing in the context of porous materials, this should be changed.
- I checked both references mentioned above: ref 12 illustrates the porosity of the deposited material by gas adsorption, ref 13 performs a density check with XRR to verify it is equal to the intended MOF material. In the present work, the authors do neither (it is indicated that the coatings are too rough for XRR). I think this is an extremely important point and I urge the authors to perform additional experiments.
- It is not clear to me why two QCM sensors are being used? Please include this in the text. How can be assured that the sensor in front of the reaction chamber is at the right temperature? A schematic of the deposition reactor would be informative.
- How should the reader come to the following conclusion? "... show self-limiting growth for both types of precursors, with some physisorption of the 1,4-BDC". I find that several statements in the manuscript are not fully supported by data, or should be much better explained.
- The reaction schemes are extremely confusing without integer numbers. I suggest the authors change this to include a Zr_6 cluster, which is of structural relevance in the UiO lattice
- It is stated that the non-saturative uptake of water is due to the presence of porosity. If the film was indeed porous, the uptake would saturate at the equilibrium uptake capacity. On the contrary to reversible uptake, the authors show themselves that reaction with water occurs (for instance in Fig 2). This section is very confusing and contains wrong conclusions in my opinion. I suggest that the authors perform a long H_2O pulse experiment to clarify this behavior. This is a crucial experiment.
- In relation to the statement 'We can at this stage only speculate that they are added through the subsequent reactions with air and acetic acid': at this point in the paper it seems more likely there are no clusters formed yet?

- Descriptions such as 'a few drops of acetic acid' are not acceptable in view of reproducibility.
- There is no information on the final thickness of the film? Also the final roughness is not quantified?
- What data / literature supports the following claim? "For purely bidentate coordination, straight columns of zirconium and 1,4-BDC linkers orientate in a denser manner than for a combination of bidentate and monodentate coordination where an expected angled coordination between 1,4-BDC and zirconium occurs."
- References:
 - o The concept of converting an amorphous UiO-66 precursor to crystalline UiO-66 has already been demonstrated and the following reference should be included: Chem. Commun. 52, 2133-2136, 2016
 - o I suggest the authors include other recent relevant work on UiO-66 deposition methods: Journal of the American Chemical Society 137, 6999-7002, 2015, Chem. Mater. 27, 1801-1807, 2015, etc.

Reviewer #2 (Remarks to the Author):

While not the first report of vapor-phase delivery of reactants for MOF film formation or even the first to employ an ALD-like method, this paper does add usefully to the MOF literature and could turn out to be a stepping stone to a broad range of MOF film syntheses. To my knowledge no one has been able to synthesize zirconium-cluster based MOFs in layer-by-layer fashion, with or w/o solvent. The Zr-O bonds are unusually strong, so this makes the material synthesized here a nontrivial test case. The paper is, in my opinion, of suitably high quality and originality to fit well with Nature Communications. One point that needs to be addressed further is the claim that the synthesized films are pinhole free. I don't doubt that in their "as synthesized" amorphous form the films are pinhole free. But after conversion to polycrystalline form it may well be that the film does contain pinholes. Perhaps the easiest way to check would be to use one side of the gold-covered QCM crystal as a working electrode in a standard three-electrode electrochemical cell in contact with a solution containing inert electrolyte and a redox active molecule that this large to fit through the MOF apertures, even if some of the linkers are missing and the apertures are larger than ideal. Commercially available tetra-phenyl porphyrin would probably be large enough. Because diffusion to pinholes is hemispherical (and therefore much more efficient than one-dimensional diffusion) redox signals turn out to be surprisingly good for detecting the presence of small number of pinholes in an electrode-supporting thin film... All-in-all, a nice paper (although I would hesitate to try the same experiment in my own lab as HCl vapor at elevated temperature is fairly corrosive). Perhaps in a follow-up study a zirconium precursor that does not release chloride could be used.

Reviewer #3 (Remarks to the Author):

The authors describe a hybrid process based on gas-phase ALD/MLD techniques to derive amorphous Zr-oxo/hydroxo terephthalate thin films which were post-deposition crystallized by annealing and exposure to acetic acid vapor in an autoclave. The presented characterization data (GIXRD, Fig. 4b and Fig 5; FTIR, Fig.6) suggest the UiO-66 phase as the major component of the finally obtained thin film material. In situ monitoring the growth by QCM with or without additional pulses with acetic acid vapor (Fig 1 and Fig 3) supports the claim of "modulation" the (self-terminated) ALD/MLD growth scheme of a more stoichiometric (amorphous) precursor thin film for the target UiO-66 material.

Solvent-free, all-gas-phase deposition/synthesis of crystalline, functional (and robust) MOF thin films are highly desirable for allowing MOF materials to be integrated to the microelectronic (thin film) device technology. Few attempts have been made, recently, using ALD/MLD techniques to obtain MOF thin films (which papers the authors cite, see Refs 12-14). The evaluation of the presented new data with this previous work does, however, not provide evidence for a highly original, new idea/approach or substantially improved results with respect to the very important over-all goal. Herein, the most relevant aspect of the new work is the choice of acetic acid as "modulator" for controlling the stoichiometry and the binding mode of the BDC linker and for initiating nucleation and crystallization of the amorphous precursor-film. However, there is no conclusive mechanism suggested or supported by the presented data which clearly makes a distinction between various forms of post deposition treatment of amorphous coordination polymer thin films to initiate crystallization. For example, a quite similar protocol for obtaining crystalline MOF-5 films after ALD/MLD deposition of an amorphous precursor film is given in Ref 14: "The as-deposited films were crystallized for 12 h in a humidity-controlled chamber at a relative humidity of 60% followed by recrystallization with DMF in an autoclave (Parr 4744) at 150 C for 120 min." What exactly is the fundamental, conceptual difference of using acetic acid rather than vapors of H₂O and DMF as modulators/initiators? It seems that each MOF system may or may not need a special additive or modulator to allow the conversion of the primarily deposited amorphous film into the targeted crystalline phase. In fact, the results documented in Ref. 12 on ALD/MLD of a MOF-2 type Cu(II)terephthalate more directly support the claim of a successful all-gas-phase synthesis of a crystalline MOF (as compared to the present manuscript). Finally, the conversion of gas-phase pre-deposited metal-oxide ultra thin films (via established ALD) into MOFs (e.g. ZIF-8) by a gas-phase treatment with the organic linker as described in Ref 12, appears to be far more developed and immediately useful for device integration and also being suitable for transfer to other MOF systems than the results presented by the authors, herein. The data and arguments provided by the authors, why their approach represents a novel and important alternative concept of MOF thin film growth are not convincing at this point.

Very important issues of MOF thin film deposition (irrespective of the technique) are phase purity (crystallinity) and(!) porosity (matching with bulk reference materials). More or less matching growth stoichiometry (mass balance) and GIXRD data are not sufficient to claim a break-through. The authors should provide quantitative (specific) porosity data with proper comparisons and rule out substantial pore blocking and/or residual amorphous components of the annealed films. Also, the authors should improve the morphological characterization of the films (SEM cross sections, quantitative roughness data). The manuscript may gain further quality if redundant information/discussion is avoided (see pp 13-14 and compare with previous sections).

In summary, the claim of a transferable concept for all-gas-phase ALD/MLD growth of functional MOFs presented and discussed in this manuscript is not sufficiently substantiated by the new data and the results seem to be limited to the specific case of UiO-66 and a special post-deposition treatment.

Rejection of the manuscript is recommended. The authors may consider submission to a more specialized journal and expansion to a full paper with more comprehensive characterization data.

Response to referees

We would like to thank the referees for their thorough work and for many constructive comments that have clearly lifted the quality of this work. We have tried to carefully follow the suggestions and believe we have been able to successfully respond to the comments below.

Kind regards

Kristian Blindheim Lausund and Ola Nilsen

Reviewer #1:

Reviewer #1 (Remarks to the Author):

The authors report a new all-vapor method to deposit the MOF material UiO-66. While I do think the work has great potential, I cannot recommend the work for publication in its current form. I would like to invite the authors to address the following issues:

- As a general comment, the manuscript is not well-structured and contains too many spelling and grammar errors to be acceptable at this level.

We have made an effort to improve this including suggestions from more colleagues.

- The authors repeatedly claim the context of microelectronics. It seems a bit 'too easy' to make such a general claim. What are the promising applications of the material UiO-66 in this context? This point should be made very clear. I urge the authors to describe their vision of what 'microelectronics' would be enabled by the described process.

We have included a reference to a recent work by Allendorf et al that describes many of the possible applications of for thin films of MOF materials such as sensors of cantilevers coated with a MOF thin films that increases the sensitivity by many orders of magnitude due to the large adsorption capabilities of the MOF, or as low-k material in electronics to reduce signal loss (Allendorf et al.).

- The authors state that solvents 'typically cause corrosion or chemical contamination in the circuitry'. I do not agree at all. Solvents do not cause corrosion. On the contrary, the reactants used ($ZrCl_4$) and products formed (HCl) would likely destroy any conductive contacts that would be present in a microelectronic system. This is a critical point that should be addressed

This has been addressed. We agree with the reviewer that the statement was somewhat unbalanced and have removed it.

- The authors state: "Possibilities for deposition of dense MOF thin films by vapour phase techniques have recently emerged" and insert ref 12 and 13. I find the term 'dense' extremely confusing in the context of porous materials, this should be changed.

This has been changed from "dense MOF thin films" to "crystalline MOF thin films" in order to avoid confusion.

- I checked both references mentioned above: ref 12 illustrates the porosity of the deposited material by gas adsorption, ref 13 performs a density check with XRR to verify it is equal to the intended MOF material. In the present work, the authors do neither (it is indicated that the coatings are too rough for XRR). I think this is an extremely important point and I urge the authors to perform additional experiments.

We have conducted further experiments to obtain information on its porosity. We have quantified the roughness of the post deposition treatment in order to explain why we are unable to perform XRR measurements. In addition, a porosity test has been performed using QCM-crystals to register the mass increase due to adsorbed water on uncoated crystals, an as deposited film, and a post deposition treated film during a 5 second water pulse in the ALD reactor. The experiment is done at room temperature and with a base pressure of 5 mbar N₂. This test shows that the films absorb a significant amount of water (more than 200 times as much as the uncoated surface for film with an approximate thicknesses of 230 nm and 500 nm respectively for as deposited and post deposition treated samples). Such a mass increase is only possible if the material is porous. We also show that the pores are more accessible in the post deposition treated sample. We have tried to quantify the amount of pores in the film based on the amount of water absorbed. The obtained number is relatively uncertain given the used technique, however neither XRR nor ellipsometric porosimetry was able to provide a result due to the surface roughness of the film.

- It is not clear to me why two QCM sensors are being used? Please include this in the text. How can be assured that the sensor in front of the reaction chamber is at the right temperature? A schematic of the deposition reactor would be informative.

The explanation of this in the methods section has been improved and relates to both increased statistics and the possibility to follow the spatial evolution of film growth in the reaction chamber.

- How should the reader come to the following conclusion? "... show self-limiting growth for both types of precursors, with some physisorption of the 1,4-BDC". I find that several statements in the manuscript are not fully supported by data, or should be much better explained.

This statement has been removed to avoid confusion. Other similar statements have been made clearer.

- The reaction schemes are extremely confusing without integer numbers. I suggest the authors change this to include a Zr₆ cluster, which is of structural relevance in the UiO lattice

Although we agree that Zr₆ clusters would be convenient for writing more comprehensible reaction schemes we have decided not to change this. The primary reason for this is that we have no evidence supporting that these clusters are formed at this stage in the process. We have therefore written the schemes that we believe fit best with the actual reactions during every ALD pulse.

- It is stated that the non-saturative uptake of water is due to the presence of porosity. If the film was indeed porous, the uptake would saturate at the equilibrium uptake capacity. On the contrary to reversible uptake, the authors show themselves that reaction with water occurs (for instance in Fig 2). This section is very confusing and contains wrong conclusions in my opinion. I suggest that the authors perform a long H₂O pulse experiment to clarify this behavior. This is a crucial experiment.

The reaction with water that the reviewer refers to (in Figure 2) is a behavior that is seen for unmodulated samples when exposed to moist air for a long period of time. The excess 1,4-BDC precursor crystallizes on the surface. When a water pulse is introduced in the ALD process we do not see any mass loss, indicating that this does not happen at that time scale. The porosity test that is mentioned above (which is conducted by a long H₂O pulse as the reviewer suggests) supports the idea that the films are porous to small molecules such as water. The reason why saturation is not seen in the ALD process is that the temperature is too high and the pulse is far too short.

- In relation to the statement 'We can at this stage only speculate that they are added through the subsequent reactions with air and acetic acid': at this point in the paper it seems more likely there are no clusters formed yet?

We agree. We believe that the clusters are not formed until the post deposition treatment in acetic acid vapor in an autoclave. We have therefore not elaborated very much on this in order not to overinterpret the results.

- Descriptions such as 'a few drops of acetic acid' are not acceptable in view of reproducibility.

This has been changed. In all cases where acetic acid was used, three drops were added. This has now been specified.

- There is no information on the final thickness of the film? Also the final roughness is not quantified?

The final thickness is now found through cross section SEM imaging. The final roughness has also been found through AFM measurements.

- What data / literature supports the following claim? "For purely bidentate coordination, straight columns of zirconium and 1,4-BDC linkers orientate in a denser manner than for a combination of bidentate and monodentate coordination where an expected angled coordination between 1,4-BDC and zirconium occurs."

We have found through XRR and QCM measurements that the density increases with modulation. Our best explanation for this is the one written above. We have, however, moderated the statement somewhat.

- References:

- o The concept of converting an amorphous UiO-66 precursor to crystalline UiO-66 has already been demonstrated and the following reference should be included: Chem. Commun. 52, 2133-2136, 2016

- o I suggest the authors include other recent relevant work on UiO-66 deposition methods: Journal of the American Chemical Society 137, 6999-7002, 2015, Chem. Mater. 27, 1801-1807, 2015, etc

The suggested references have been added.

Reviewer #2:

- While not the first report of vapor-phase delivery of reactants for MOF film formation or even the first to employ an ALD-like method, this paper does add usefully to the MOF literature and could turn out to be a stepping stone to a broad range of MOF film syntheses. To my knowledge no one has been able to synthesize zirconium-cluster based MOFs in layer-by-layer fashion, with or w/o solvent. The Zr-O bonds are unusually strong, so this makes the material synthesized here a nontrivial test case. The paper is, in my opinion, of suitably high quality and originality to fit well with Nature Communications. One point that needs to be addressed further is the claim that the synthesized films are pinhole free. I don't doubt that in their "as synthesized" amorphous form the films are pinhole free. But after conversion to polycrystalline form it may well be that the film does contain pinholes. Perhaps the easiest way to check would be to use one side of the gold-covered QCM crystal as a working electrode in a standard three-electrode electrochemical cell in contact with a solution containing inert electrolyte and a redox active molecule that this large to fit through the MOF apertures, even if some of the linkers are missing and the apertures are larger than ideal. Commercially available tetra-phenyl porphyrin would probably be large enough. Because diffusion to pinholes is hemispherical (and therefore much more efficient than one-dimensional diffusion) redox signals turn out to be surprisingly good for detecting the presence of small number of pinholes in an electrode-supporting thin film... All-in-all, a nice paper (although I would hesitate to try the same experiment in my own lab as HCl vapor at elevated temperature is fairly corrosive). Perhaps in a follow-up study a zirconium precursor that does not release chloride could be used.

Firstly, we would like to thank you for your kind words regarding the quality of our work. As you mentioned, the ZrO bonds are very strong which makes this an interesting and nontrivial test case. This is an important point, and we appreciate that you address this.

Our depositions have not been done in clean room conditions hence we have not claimed that they are pinhole free. We have only stated that the ALD technique in general has this advantage. In order to avoid confusion, these statements have been removed. We do, however, plan to address this in further works, and thank you for your good suggestions for a testing procedure.

Reviewer #3:

- The authors describe a hybrid process based on gas-phase ALD/MLD techniques to derive amorphous Zr-oxo/hydroxo terephthalate thin films which were post-deposition crystallized by annealing and exposure to acetic acid vapor in an autoclave. The presented characterization data (GIXRD, Fig. 4b and Fig 5; FTIR, Fig.6) suggest the UiO-66 phase as the major component of the finally obtained thin film material. In situ monitoring the growth by QCM with or without additional pulses with acetic acid vapor (Fig 1 and Fig 3) supports the claim of "modulation" the (self-terminated) ALD/MLD growth scheme of a more stoichiometric (amorphous) precursor thin film for the target UiO-66 material.
- Solvent-free, all-gas-phase deposition/synthesis of crystalline, functional (and robust) MOF thin films are highly desirable for allowing MOF materials to be integrated to the microelectronic (thin film) device technology. Few attempts have been made, recently, using ALD/MLD techniques to obtain MOF thin films (which papers the authors cite, see Refs 12-14). The evaluation of the presented new data with this previous work does, however, not provide evidence for a highly original, new idea/approach or substantially improved results with respect to the very important over-all goal. Herein, the most relevant aspect of the new work is the choice of acetic acid as "modulator" for controlling the stoichiometry and the binding mode of the BDC linker and for initiating nucleation and crystallization of the amorphous precursor-film. However, there is no conclusive mechanism suggested or supported by the presented data which clearly makes a distinction between various forms of post deposition treatment of amorphous coordination polymer thin films to initiate crystallization. For example, a quite similar protocol for obtaining crystalline MOF-5 films after ALD&MLD deposition of an amorphous precursor film is given in Ref 14: "The as-deposited films were crystallized for 12 h in a humidity-controlled chamber at a relative humidity of 60% followed by recrystallization with DMF in an autoclave (Parr 4744) at 150 C for 120 min." What exactly is the fundamental, conceptual difference of using acetic acid rather than vapors of H₂O and DMF as modulators/initiators?

We thank the reviewer for a thorough comparison of different types of processes. We would argue that there are some distinct differences that makes our work stand out in this comparison. Sami et al. use a two-step crystallization process where the films are exposed to moisture and then solvothermally treated in DMF corresponding to the traditional bulk synthesis of MOFs. With our process we have eliminated the solvent DMF which is toxic and which would cause stiction of small components. Furthermore, Salmi et al. have used their process to make MOF-5 which is a less stable and less significantly important MOF material than UiO-66. MOF-5 is known to be unstable towards water whereas UiO-66 is considered to be very stable, consisting of Zr₆ clusters. A process for production of UiO-66 thin films therefore opens for many more applications than MOF-5 thin films, such as integration in aqueous environments. As the reviewer points at in the beginning, this has been possible only when using modulation of the deposited composition by additional acetic acid pulses.

- It seems that each MOF system may or may not need a special additive or modulator to allow the conversion of the primarily deposited amorphous film into the targeted crystalline phase. In fact, the results documented in Ref. 12 on ALD/MLD of a MOF-2 type Cu(II)terephthalate more directly support the claim of a successful all-gas-phase synthesis of a crystalline MOF (as compared to the present manuscript).

It is true that the process for formation of MOF-2 Cu(II)terephthalate is more direct in that it does not need a post deposition treatment. However, this is a less complex structure without metal clusters. It should also be noted that this MOF-2 type structure is less stable than UiO-66 and far less porous which makes it less desirable for applications.

- Finally, the conversion of gas-phase pre-deposited metal-oxide ultra thin films (via established ALD) into MOFs (e.g. ZIF-8) by a gas-phase treatment with the organic linker as described in Ref 12, appears to be far more developed and immediately useful for device integration and also being suitable for transfer to other MOF systems than the results presented by the authors, herein. The data and arguments provided by the authors, why their approach represents a novel and important alternative concept of MOF thin film growth are not convincing at this point.

It is true that Stassen et al. have demonstrated patterning of their MOF. However, this patterning is not a fundamental part of the synthesis process but rather a demonstration of what a deposition technique such as ALD or CVD can be used for. We have not patterned our films yet, but we have no doubts that this is achievable. When the reviewer states that it appears to be more “immediately useful for device integration and also being suitable for transfer to other MOF systems” we are not sure that we agree and believe this to be a question that the future will answer. We would again like to emphasize the higher applicability of the more stable and more complex UiO-66 structure. With regards to the comment that the Stassen process is suitable for transfer to other MOF systems, we highly doubt that this is suitable for the UiO-66 structure due to its complexity.

- Very important issues of MOF thin film deposition (irrespective of the technique) are phase purity (crystallinity) and(!) porosity (matching with bulk reference materials). More or less matching growth stoichiometry (mass balance) and GIXRD data are not sufficient to claim a break-through. The authors should provide quantitative (specific) porosity data with proper comparisons and rule out substantial pore blocking and/or residual amorphous components of the annealed films. Also, the authors should improve the morphological characterization of the films (SEM cross sections, quantitative roughness data). The manuscript may gain further quality if redundant information/discussion is avoided (see pp 13-14 and compare with previous sections).

As mentioned in the reply to reviewer #1, a porosity test has been conducted using QCM measurements to determine the amount of adsorbed water to uncoated crystals (control), an as deposited film, and a post deposition treated film. This test shows that the films are indeed porous

and absorb more than 200 times as much water than an uncoated surface under vacuum conditions at room temperature. We have also now included SEM cross section images and AFM images to highlight the morphology of the deposited and conformed films.

- In summary, the claim of a transferable concept for all-gas-phase ALD/MLD growth of functional MOFs presented and discussed in this manuscript is not sufficiently substantiated by the new data and the results seem to be limited to the specific case of UiO-66 and a special post-deposition treatment.
- Rejection of the manuscript is recommended. The authors may consider submission to a more specialized journal and expansion to a full paper with more comprehensive characterization data.

Reviewers' comments:

Reviewer #1 (Remarks to the Author):

I appreciate the efforts made by the authors to improve the quality of the manuscript. I think this work would form a valuable contribution to both the MOF and ALD communities. However, the manuscript is not ready for publication before the following issues have been addressed:

- Several spelling and grammar errors remain ("gracing incidence" XRD, "an uncoated crystals", "FTIR spectre", "J. A. Wollam", ...). I do not feel it is my job as a reviewer to point out every single spelling mistake in the manuscript. In my opinion the authors could have put in more effort, especially as I pointed this out already in the previous report.
- The section on water uptake by the as-deposited film is still not clear. It is stated that "water reacts considerably with the film" but also that the uptake is an indication that "the film is porous to small molecules". Which one is it? The manuscript cannot be published as long as there is no clarity on this. Also, the statement that water reacts with the film is not commensurate with the statement (in the same paragraph!) that the Zr-carboxylate bond is stable. Please do the relevant experiments and present a clear picture.
- Include the water uptake curve by the as-deposited film (overlaid with the AA-treated film) to show the saturation behavior. If not in the text, then in the SI.
- I appreciate the water uptake experiment for the crystallized UiO coatings. However, the authors should extract more data from this experiment. I specifically want to know what the wt% water uptake in the MOF is and if this matches with literature data under similar conditions. In case no matching literature data is available, it should be easy enough to redo the same experiment with QCM crystals coated with solvothermally grown MOF crystallites.
- In the scenario of reaction with water, uptake in the film would of course also saturate. To me it feels that the Zr-carboxylate bonds partially hydrolyze and that this bond is not stable (as the authors claim). The reaction case seems to match with the statement that "a large mass increase for the H₂O pulse is observed, a majority of it is lost during the subsequent purge, provided this is sufficiently long". In the case of only adsorption in pores, all water should be removed (after waiting long enough or increasing the substrate temperature). Please include FTIR spectra before and after water exposure.
- The Zr ions bind to a carboxylate, not a protonated carboxylic acid. Make the necessary adjustments to the text.
- I understand the use of two QCM sensors. However, it is not clear to me how the reader should come to the following conclusion: "delayed saturation of ZrCl₄ along the flow stream". This conclusion is made based on one data point? Also, it is not clear why two red and two black data sets seem to be plotted on top of each other? Why two of each?
- Please include how FTIR spectra were recorded? In transmission?

Reviewer #3 (Remarks to the Author):

The revised and updated manuscript has been improved according to the three reviewers`

suggestions and criticisms. Nevertheless, this reviewer is still not (fully) convinced by the arguments of the authors that this work is really original and novel in the sense outlined in the previous review report. As indicated before, the described process is not essentially a full CVD/ALD type deposition of a MOF. It is a hybrid process in which an amorphous material (the precursor of the final MOF) deposited by CVD/ALD is transformed in a post deposition step by an autoclave (annealing type) gas/solid reaction (which is not a deposition). I still reason, that the title and the central claim which the title implies is not fully substantiated by the presented data.

Anyway: Some technical issues remain and need to be considered and fixed. The presentation and discussion of the results after the autoclave reactions are convincing, however, still, the results before autoclave treatment need to be evaluated in a more thorough way. The authors may consider consulting with the experts in coordination chemistry, probably preferably from MOF community. The authors should consider the following issues and clarify precisely.

- 1) In line 100 of the manuscript, the authors claimed that the $ZrCl_4$ -H2bdc amorphous film is porous based on the loss of mass due to the possible desorption of water. If we accept this claim, the authors did not address the issue of loss of HCl as by-product, and decay in chloride content in the elemental content from their experimental results as given in the equation at lines 86 and 87. This loss of mass in the presence of water is highly due to loss of HCl upon reaction of the physical mixture/hybrid composite of $ZrCl_4$ -H2bdc and may be along with the loss of excess water. So the authors claim of porosity in this paragraph is not convincing.
- 2) The corrosive nature of HCl by product for the fabrication of microelectric applications is not yet being properly addressed in the manuscript which was raised by the both reviewers 1 and 2.
- 3) Figure 2d and the results referring to the respective thin film material need further explanation. The visual conclusion from SEM images is that the amorphous $ZrCl_4$ -H2bdc composite converted into crystalline films with very large non-uniform crystallites of pure H2bdc and might be along with a dechlorinated Zr-bdc composite material. By looking at the crystallite shapes of the material, how significant are the XRR measurements and the results of this film, which was mentioned in line 126?
- 4) In line 273: The authors suggested the presence of crystalline H2bdc in the films (Figure 7, blue line), according to FTIR data. The authors should to measure and provide the FTIR spectra of pure H2bdc also for comparison (for SI data)
- 5) Three drops of acetic acid are still imprecise. The exact quantity in microliters or millilitres should be given.
- 6) There are still few typos and mistakes existing in the manuscript. For example, in line 48, 'trough' instead of 'through' and in line 283 'methode' instead of 'method'.

With another revision which eliminates possible misconceptions and rigorous proof reading the manuscript may well be acceptable for Nature Communications.

Response to the reviewers

We would again like to thank the reviewers for their good work, and appreciate the level of detail in their response. As a consequence of the comment raised by the reviewers, we believe that our explanation of what happens during the deposition with water pulsing paints a much clearer picture now.

Quite a few changes have been made in the manuscript, both to address the comments by the reviewer and the points in the checklist provided by the editor. All changes are highlighted on the request of the editor. Changes, additions or deletions are written in blue text, and paragraphs that have been moved in the manuscript are written in green text.

A point by point response to the issues raised by the reviewers is included below.

Kind regards

Kristian Lausund and Ola Nilsen

Reviewers' comments:

Reviewer #1 (Remarks to the Author):

I appreciate the efforts made by the authors to improve the quality of the manuscript. I think this work would form a valuable contribution to both the MOF and ALD communities. However, the manuscript is not ready for publication before the following issues have been addressed:

- Several spelling and grammar errors remain ("gracing incidence" XRD, "an uncoated crystals", "FTIR spectre", "J. A. Wollam", ...). I do not feel it is my job as a reviewer to point out every single spelling mistake in the manuscript. In my opinion the authors could have put in more effort, especially as I pointed this out already in the previous report.

A much more thorough proof reading has been done both by us and a by a British colleague.

- The section on water uptake by the as-deposited film is still not clear. It is stated that "water reacts considerably with the film" but also that the uptake is an indication that "the film is porous to small molecules". Which one is it? The manuscript cannot be published as long as there is no clarity on this. Also, the statement that water reacts with the film is not commensurate with the statement (in the same paragraph!) that the Zr-carboxylate bond is stable. Please do the relevant experiments and present a clear picture.

This section has been changed in order to clarify what happens. We believe that when water is pulsed it removes the excess 1,4-BDC (much in the same way as acetic acid). We have added FTIR and GIXRD results that support this in the supplementary information.

However, we still think that the films are porous to small molecules like water. There is a large mass increase when water is pulsed. We believe this is due to water adsorbing in the pores (a similar increase is not seen when acetic acid is pulsed). During the water pulse, we believe that there is also a small mass loss due to release of Cl and 1,4-BDC. This loss is concealed by the large mass increase from adsorbing water. We believe that the large mass loss observed during the purge following this pulse is water leaving the pores. The claim that the films are porous is also supported by the porosity test in Figure 8.

- Include the water uptake curve by the as-deposited film (overlaid with the AA-treated film) to show the saturation behavior. If not in the text, then in the SI.

This has been added to the supplementary information

- I appreciate the water uptake experiment for the crystallized UiO coatings. However, the authors should extract more data from this experiment. I specifically want to know what the wt% water uptake in the MOF is and if this matches with literature data under similar conditions. In case no matching literature data is available, it should be easy enough to redo the same experiment with QCM crystals coated with solvothermally grown MOF crystallites.

We have calculated the wt% water uptake (1.9 wt% with a relative humidity in the ALD reactor of ~7 %), and it correspond well with literature data (~2 wt% at a relative humidity in the same range). This has been included in the text.

- In the scenario of reaction with water, uptake in the film would of course also saturate. To me it feels that the Zr-carboxylate bonds partially hydrolyze and that this bond is not stable (as the authors claim). The reaction case seems to match with the statement that "a large mass increase for the H₂O pulse is observed, a majority of it is lost during the subsequent purge, provided this is sufficiently long". In the case of only adsorption in pores, all water should be removed (after waiting long enough or increasing the substrate temperature). Please include FTIR spectra before and after water exposure.

As mentioned above, we now think that the excess, monodentate 1,4-BDC is removed when water is pulsed. In other words, this bond is not stable. This is supported by the FTIR and GIXRD results in Supplementary Figure 1.

From the pulsing scheme in Figure 4, it is difficult to say whether all of the mass of adsorbed water is lost, because some of the mass loss is due to loss of 1,4-BDC and Cl. However, we have made an attempt at describing this in more detail, and have added that it seems like the mass loss in the water purge carries over to the purges following the ZrCl₄ and 1,4-BDC pulses, giving a significant mass loss also in these purges, which was not seen in the deposition without water. If we take this mass loss into account, the total mass loss from the start of the water purge to the end of the 1,4-BDC purge is more than the mass increase when water is pulsed. This corresponds well with the current claim that we have a loss of 1,4-BDC that is concealed by the mass increase from water adsorption during the water pulse, followed by a mass loss due to water desorption in the water purge (and in the ZrCl₄ and 1,4-BDC pulses and purges). As mentioned in the text, this may cause unwanted reactions between water and ZrCl₄, which is why we have not gone further with this method.

However, even though the monodentate bond is unstable towards water, the bidentate 1,4-BDC bond is stable. This is supported by the FTIR results in Supplementary figure 3. This is the coordination that is found in the modulated, as deposited sample that is relevant for the porosity test in Figure 8. The bridging bond that is found in the post deposition treated sample (that is also relevant for the porosity test) should also be stable towards water based on the reported stability of UiO-66 in literature (for instance, Leus et al.¹ report that UiO-66 is stable in water for 2 months and also stable in steam at 200 °C for 5h). In other words, the films do not react with water during the porosity test, which is shown in Figure 8.

- The Zr ions bind to a carboxylate, not a protonated carboxylic acid. Make the necessary adjustments to the text.

This has been changed.

- I understand the use of two QCM sensors. However, it is not clear to me how the reader should come to the following conclusion: "delayed saturation of ZrCl₄ along the flow stream". This conclusion is made based on one data point? Also, it is not clear why two red and two black data sets seem to be plotted on top of each other? Why two of each?

This is based on the fact that we see a lower growth rate in the back (downstream) of the reaction chamber when the pulse length is 0.75, 1 and 2 seconds. The reason for the double set of the data points is that the experiment was repeated two times. This has been explained in the text.

- Please include how FTIR spectra were recorded? In transmission?

This has been included in the methods section.

Reviewer #3 (Remarks to the Author):

The revised and updated manuscript has been improved according to the three reviewers' suggestions and criticisms. Nevertheless, this reviewer is still not (fully) convinced by the arguments of the authors that this work is really original and novel in the sense outlined in the previous review report. As indicated before, the described process is not essentially a full CVD/ALD type deposition of a MOF. It is a hybrid process in which an amorphous material (the precursor of the final MOF) deposited by CVD/ALD is transformed in a post deposition step by an autoclave (annealing type) gas/solid reaction (which is not a deposition). I still reason, that the title and the central claim which the title implies is not fully substantiated by the presented data.

Our title emphasizes that this technique is an all gas process. This is true also with the post deposition treatment. Therefore, we will argue that our title does not promise more than we deliver.

It is true that our technique does not provide crystalline UiO-66 through only ALD/MLD. However, we do obtain porous films as deposited that can be further crystallised into UiO-66 through a simple post deposition treatment.

As for the novelty of this work, we would still like to point out that the thermal and chemical stabilities of UiO-66 make it more suitable for applications than many other MOFs. In addition, the complexity of the UiO-66 structure (especially the clusters) sets it apart from other MOFs that have been made by ALD/MLD or CVD processes. Another important point to the novelty of this paper is the modulation with acetic acid, which controls the stoichiometry of linker to metal atoms directly in the deposition. This is likely to enable deposition of other MOFs and maybe also other complex materials with ALD, and should therefore be a valuable addition to the field.

Anyway: Some technical issues remain and need to be considered and fixed. The presentation and discussion of the results after the autoclave reactions are convincing, however, still, the results before autoclave treatment need to be evaluated in a more thorough way. The authors may consider consulting with the experts in coordination chemistry, probably preferably from MOF community. The authors should consider the following issues and clarify precisely.

1) In line 100 of the manuscript, the authors claimed that the $ZrCl_4$ -H2bdc amorphous film is porous based on the loss of mass due to the possible desorption of water. If we accept this claim, the authors did not address the issue of loss of HCl as by-product, and decay in chloride content in the elemental content from their experimental results as given in the equation at lines 86 and 87. This loss of mass in the presence of water is highly due to loss of HCl upon reaction of the physical mixture/hybrid composite of $ZrCl_4$ -H2bdc and may be along with the loss of excess water. So the authors claim of porosity in this paragraph is not convincing.

The loss of Cl in the presence of water has been taken into account. However, we still believe that the films are porous.

There is a large mass increase when water is pulsed. We believe this is due to water adsorbing in the pores (a similar increase is not seen when acetic acid is pulsed). During the water pulse, we believe that there is also a small mass loss due to release of Cl and 1,4-BDC. This loss is concealed by the large mass increase from adsorbing water. We believe that the large mass loss observed during the purge following this pulse is water leaving the pores. The claim that the films are porous is also supported by the porosity test in Figure 8.

2) The corrosive nature of HCl by product for the fabrication of microelectric applications is not yet being properly addressed in the manuscript which was raised by the both reviewers 1 and 2.

We agree that HCl is corrosive, and we have previously removed the claim that corrosion is prevented by this technique. However, we still think this is a very useful technique for microelectronics because it is an all-gas process, which means that stiction of small components, will be avoided.

3) Figure 2d and the results referring to the respective thin film material need further explanation. The visual conclusion from SEM images is that the amorphous ZrCl₄-H₂bdc composite converted into crystalline films with very large non-uniform crystallites of pure H₂bdc and might be along with a dechlorinated Zr-bdc composite material. By looking at the crystallite shapes of the material, how significant are the XRR measurements and the results of this film, which was mentioned in line 126?

This seems to be a misunderstanding. The XRR measurements that were mentioned in line 126 (now in line 125) refer to the as deposited film. They are flat, as can be seen in Figure 3c (previously Figure 2c). The film in Figure 3d (previously Figure 2d) is the film that has been exposed to a relative humidity of 70-70 % for 24h. This is, as you correctly point out, rough and consisting of pure 1,4-BDC crystallites along with the Zr-BDC hybrid film. In order to avoid this misunderstanding, the reference to the figure in the text has been made clearer.

4) In line 273: The authors suggested the presence of crystalline H₂bdc in the films (Figure 7, blue line), according to FTIR data. The authors should to measure and provide the FTIR spectra of pure H₂bdc also for comparison (for SI data)

This FTIR measurement has been done, and it is clear that the broad peak 3207 cm⁻¹ does not correspond to the spectrum for crystalline 1,4-BDC.

We have looked into this further, and our best suggestion is that the broad peak 3207 cm⁻¹ is due to OH in carboxylic acid groups on monodentate 1,4-BDC that only coordinates to one Zr atom. This agrees well with the conclusions that we can draw from the splitting of the peaks around 1500 cm⁻¹ where we see that the sample that shows this broad peak should have monodentate 1,4-BDC molecules whereas the two other samples that do not have this broad peak do not have monodentate 1,4-BDC molecules.

This has been changed in the text. (The FTIR spectrum for 1,4-BDC has not been included since it is not relevant for our current claim)

5) Three drops of acetic acid are still imprecise. The exact quantity in microliters or millilitres should be given.

This has been changed to ca. 0.1 ml.

6) There are still few typos and mistakes existing in the manuscript. For example, in line 48, 'trough' instead of 'through' and in line 283 'methode' instead of 'method'.

As mentioned above, a much more thorough proof reading has been done both by us and by a British colleague.

With another revision which eliminates possible misconceptions and rigorous proof reading the manuscript may well be acceptable for Nature Communications.

Reference:

- 1 Leus, K. *et al.* Systematic study of the chemical and hydrothermal stability of selected “stable” Metal Organic Frameworks. *Microporous and Mesoporous Materials* **226**, 110-116, doi:<http://dx.doi.org/10.1016/j.micromeso.2015.11.055> (2016).

REVIEWERS' COMMENTS:

Reviewer #1 (Remarks to the Author):

The Authors adequately addressed almost all comments and I recommend the publication of the manuscript in Nature Communications after the last point has been addressed:

Please include in the porosity test (Fig 8) a demonstration that the QCM returns to baseline after purging long enough. This experiment (and cycling of adsorption-desorption) would be a much stronger proof of the porosity of the as-deposited film than what is currently given, as reactive water uptake is excluded that way.

Still some spelling errors (incl. in Supporting Information): 'acidic acid' (should be 'acetic acid'), 'FTIR spectres',...

Reviewer #3 (Remarks to the Author):

The authors addressed all the remaining issues and questions sufficiently. The manuscript has gained quite a lot during the revision process. I am happy to recommend publication of this important work in Nature Communications.

[The only remaining disagreement may refer to the wording of the title and the immediate connection of "all gas-phase" with ALD, without emphasizing the essential two step process of first depositing an amorphous - not nucleated - "precursor-film", which is then crystallized in a separate gas-phase, which second step, however, is not an ALD process. Well, since this important distinction is given in the abstract and the discussion of this issue has improved, I agree with the authors that their title - essentially - does not promise more, than their work delivers.]

Response to referees

We would again like to thank the referees for their thorough work through all the stages of the review process and for the many constructive comments that have lifted the quality of this work. We here answer the final remarks by the referees:

REVIEWERS' COMMENTS:

Reviewer #1 (Remarks to the Author):

The Authors adequately addressed almost all comments and I recommend the publication of the manuscript in Nature Communications after the last point has been addressed:

Please include in the porosity test (Fig 8) a demonstration that the QCM returns to baseline after purging long enough. This experiment (and cycling of adsorption-desorption) would be a much stronger proof of the porosity of the as-deposited film than what is currently given, as reactive water uptake is excluded that way.

A demonstration that the QCM returns to the baseline after a sufficiently long purge is provided in Supplementary Figure 2b.

Still some spelling errors (incl. in Supporting Information): 'acidic acid' (should be 'acetic acid'), 'FTIR spectres',...

We once again apologize for not observing these errors and have again gone through the document with focus on grammar and spelling.

Reviewer #3 (Remarks to the Author):

The authors addressed all the remaining issues and questions sufficiently. The manuscript has gained quite a lot during the revision process. I am happy to recommend publication of this important work in Nature Communications.

[The only remaining disagreement may refer to the wording of the title and the immediate connection of "all gas-phase" with ALD, without emphasizing the essential two step process of first depositing an amorphous - not nucleated - "precursor-film", which is then crystallized in a separate gas-phase, which second step, however, is not an ALD process. Well, since this important distinction is given in the abstract and the discussion of this issue has improved, I agree with the authors that their title - essentially - does not promise more, than their work delivers.]

We thank you for your consideration and discussions throughout the reviewing process and feel that we have reached a consensus regarding the title.